# Integrating Linguistic, Archaeological and Genetic Perspectives Unfold the Origin of Ugrians

**DOI:** 10.3390/genes14071345

**Published:** 2023-06-26

**Authors:** Tibor Török

**Affiliations:** 1Department of Genetics, University of Szeged, H-6726 Szeged, Hungary; torokt@bio.u-szeged.hu; 2Department of Archaeogenetics, Institute of Hungarian Research, H-1041 Budapest, Hungary

**Keywords:** conquering Hungarians, Sargat, Gorokhovo, Xiongnu, Hun, Uralic, Ugric, Seima-Turbino

## Abstract

In the last year two publications shed new light on the linguistic and genomic history of ancient Uralic speakers. Here I show that these novel genetic and linguistic data are compatible with each-other and with the archaeological inferences, allowing us to formulate a very plausible hypothesis about the prehistory of Ugric speakers. Both genetic and archaeological data indicate the admixture of the Mezhovskaya population with northern forest hunters in the late Bronze Age, which gave rise to a “proto-Ugric” community. This finding is consistent with the linguistic reconstruction of the proto-Ugric language. Genetic data indicate an admixture of proto-Hungarians with early Sarmatians and early Huns, and I show that the first admixture can be reconciled with the formation of the Gorokhovo culture and its integration into the early Sarmatian Prokhorovka culture, while the second admixture corresponds to the transformation of the Sargat and Sarmatian cultures due to Xiongnu invasions.

## 1. Introduction

Since the emergence of the ancient DNA field, one of the main ambitions of researchers has been the integration of linguistic, archaeological and genetic data, in order to link language and population histories [1]. Recently, this multidisciplinary approach brought about real breakthroughs by unravelling the origin of the Indo-European [2] and Transeurasian [3] languages. The collaboration of researchers from these three different scientific fields successfully identified the homelands of ancestral languages as well as the archaeological cultures responsible for dispersing the daughter language subfamilies.

Although for the Uralic languages such a comprehensive synthesis is not yet available, last year the linguistic synthesis of Grünthal et al. [4] and the ancient DNA genome data of Maróti et al. [5] shed new light on the history of ancient Uralic speakers, which opens the door to inspect the correlation between the linguistic, genetic and archaeological inferences. The present overview is confined to the Ugric subgroup of Uralic-speakers.

According to the traditional view, established with the comparative linguistic method, the Uralic languages descended from a common proto-language, which was progressively split into daughter branches [6]. The Ugric branch consists of Hungarian and Ob-Ugric, the latter is represented by the Khanty and Mansi languages. Most linguistic analyses support the genealogical relatedness of the Ugric group, with a phylogenetic split into Hungarian and Ob-Ugric [7], while some data signify a closer link of Hungarian to Mansi [8]. A recent study revealed fairly good concordances between the genetic and linguistic distances measured between modern Uralic speakers, suggesting co-dispersion of genes and languages [9]. Uralic-speaking groups were shown to share a distinct, Nganasan-like genomic ancestry component originating from Siberia, with Hungarians being an exception. Nonetheless, we have demonstrated that this ancestry component was present in the conquering Hungarians [5], but it was diluted to a nearly undetectable level in their European environment, especially after their migration to the Carpathian Basin. This is the main reason for the significant genetic difference detected between the modern Hungarian-speaking and Ob-Ugric-speaking groups.

Here I show that the results of the three independent scientific disciplines—linguistics, genetics, and archaeology—can be fairly reconciled, enabling a rather plausible reconstruction of the major events in the prehistory of Ugric speakers. Moreover, this multidisciplinary approach also points at missing information, which should be addressed by future genetic research, as much of the relevant archaeological material has not been studied yet. 

## 2. Results

### 2.1. Population History of the Carpathian Basin

Modern Hungarians stand out as linguistically isolated in Europe, despite their genetic similarity to the surrounding populations [10,11]. This paradox calls for an explanation, as the co-dispersion of genes and languages is typically observed [12,13,14,15], even among groups speaking Uralic languages [9]. The Hungarian language is attributed to the immigrant conquering Hungarians, believed to be the descendants of a hypothetical proto-Ugric people [16]. Although historical sources report the settlement of various groups of Asian origin in the Carpathian Basin, such as Huns, Avars, Pechenegs, Jazyg people, and Cumans, in addition to the conquering Hungarians, the proportion of Asian genome components in present-day Hungarians does not exceed 4% [17]. This can be attributed to two factors. Firstly, the foundational population carrying the common European gene pool remained in a significant majority throughout the migratory periods in the Carpathian Basin [5]. Secondly, following the devastations caused by the Mongol and Turkish invasions, settlers from other parts of Europe played a significant role in establishing the modern genetic makeup of the Carpathian Basin [18]. This common European gene-pool was formed by the Bronze Age through the admixture of three sources: Western Hunter-Gatherers, the first Homo sapiens sapiens appearing in Paleolithic Europe, Neolithic farmers originating from Anatolia, and Yamnaya steppe migrants that arrived in the late Neolithic to early Bronze Age [19,20]. 

This common European gene pool, also characteristic of the Carpathian Basin, has been overlaid by migration waves originating from the east since the Iron Age. During the 9th century BCE, smaller groups of pre-Scythians (Cimmerians) of the Mezőcsát culture appeared, and the single analyzed genome from their scarce findings contained Asian elements [21]. The classic Scythian culture spread across the Great Hungarian Plain between the late 7th–early 6th century BCE, but their currently available genetic data represent the genetic profile of the local European population [22]. From 50 BCE the Sarmatians arrived in multiple waves, leaving a significant archaeological heritage behind. However, the 17 Sarmatian individuals genetically examined also appear to belong to the genetic legacy of the local European population [23]. 

Subsequently, during the Migration Period, a substantial number of peoples arrived from the East, including the Huns (400–453 AD), Avars (568–822), and the Magyars of the Hungarian Conquest (862–905 AD). The Huns left little trace during their brief presence, but the military leadership of the European Huns seems to have descended from the Asian Huns (Xiongnus), while the majority of them consisted of subjugated Germanic and Sarmatian populations [5]. The most significant influx of genes from Asia clearly occurred during the Avar period, arriving in multiple waves. The ruling elite of the Avars originated from the Rouran Empire in Mongolia [23], but a significant portion of the masses they brought in consisted of mixed-origin populations that had emerged in the Pontic-Caspian steppe during the Hunnic era [5].

The conquering Hungarians exerted the most enduring political and cultural impact on the region, as their descendants from the ruling elite established the Hungarian state. Initial studies of uniparental lineages revealed the presence of approximately 30% Eastern Eurasian components in both sexes of the immigrant elite [5,24,25,26,27], while the commoner population appears to have carried the overlaid local European gene pool from previous eastern immigrations [28]. The genome data of this population is described in the following section. 

### 2.2. Genome Data

Because of the careful sampling, large sample size and high-resolution genome analysis, the latest genetic study could reconstruct the population history of the Huns, Avars and conquering Hungarians of the Carpathian Basin with fairly great accuracy [5]. As the subject of the present survey is confined to the Hungarians who immigrated in the Carpathian Basin during the 10th–11th centuries CE, here I will give a short summary of the major findings related to this group. 

From the analysis of 113 genomes from the 10th–11th century ancient Hungarians, we could identify first generation immigrants (the actual conquerors), native residents and subsequent admixed progenies of these two groups. As expected, most of the locals, with principally European genomes, were buried in large “village” cemeteries of the common people, while most of the immigrants, with eastern genome components, were buried in the small “camp” cemeteries of the contemporary elite. Though the immigrants were assembled from genetically diverse sources, their predominant component originated from a single, genetically well-defined source, referred to as “Conqueror Asia Core” (Conq_Asia_Core), whose ancestors once lived around northern Kazakhstan, between the Ural and Altai regions. Twelve individuals carried this component in a pure form, without recent European admixture, and they belonged to the elite according to the archaeological assessment. Elite status is indicated by jewelry, clothing ornaments, partial horse burial, horse-riding equipment and weaponry. A substantial part of the studied population carried the Conq_Asia_Core component in an admixed form.

Based on F3-statistics, F4-statistics and qpAdm models, the Conq_Asia_Core had a common evolutionary history with the ancestors of Mansis, the closest language relatives of modern Hungarians, and their common ancestors belonged to a single “proto-Ugric” population. We have also shown that this proto-Ugric community originated from the admixture of the Mezhovskaya population with ancestors of modern Nganasans in the late Bronze Age. The Mezhovskaya population had been shown to derive from the Andronovo population with some East Asian admixture [19], and could be qpAdm-modelled from 74% Srubnaya, 18% Nganasan and 8% Ancient North Eurasian (ANE) components [29]. As we modelled Mansis from 48% Mezhovskaya, 44% Nganasan and 8% ANE, it follows that the proto-Ugric gene pool emerged from the further influx of the same, or very similar East Asian population in the Mezhovskaya territory that shaped the Mezhovskaya itself from the Andronovo substrate.

The Conq_Asia_Core genome could be qpAdm-modelled from 50% Mansi, 35% early Sarmatian and 15% Xiongnu/Hun ancestries. The Sarmatian admixture was dated between 643–431 BCE, while the Hun admixture was dated between 217–315 CE with the DATES algorithm. The Mansis did not take part in these admixture processes, indicating that they must have separated from the ancestors of Conq_Asia_Core before 643–431 BCE, in the early Iron Age, and in isolation conserved their proto-Ugric gene pool. I will refer to the theoretical ancestral population of Conq_Asia_Core, which separated from the Mansis as proto-Hungarian. 

### 2.3. Consistency of Genetic and Linguistic Data

Linguists have reconstructed a “common Ugric” proto-language [30], which could be spoken by our genetically reconstructed “proto-Ugric” community. The divergence of the Ugric linguistic clade into Hungarian and Ob-Ugric was estimated to have occurred between 3421–745 BCE [31] and its upper bond is within the range of an Iron Age separation inferred from genetic data.

Using linguistic, paleoclimatic and archaeological data Grünthal et al., 2022 [4] reconstructed the most likely origin and spread of the Uralic-speaking populations. The authors have taken into account and thoroughly discussed all available linguistic data and models to create their newest model that fits the data the best and is the least contradictory. They located the Uralic homeland in Western Siberia, specifically around the Minusinsk Basin. Early Uralic-speaking groups are believed to have spread relatively quickly westward with the Seima-Turbino phenomenon, which occurred in the early Bronze Age and followed the rivers in the forest-steppe zone (see Figure 1). The Seima-Turbino archaeological complex is notable for its high-quality tin–bronze artifacts, whose tin source was derived from the Altai. The authors argue that Uralic speakers played a crucial role in the transportation of metal from the Ural and Sayan mines. Furthermore, they assert that Uralic speakers actively engaged in various occupations, such as prospecting, mining, boating, and trade management, within the trading posts that were established at significant river confluences. These trading posts facilitated commerce between Uralic-speaking groups and their Indo-Iranian-speaking neighbors.

An important implication of the model is that the archaeological cultures appearing in the Seima-Turbino zone in later periods were supposedly Uralic-speaking. The original Ugric range is predicted to have been east of the Urals, south of the forest zone and not far from the steppe. This implies that the middle Bronze Age Cherkaskul culture (18th–16th centuries BCE), and its late Bronze Age successor Mezhovskaya (13th–7th centuries BCE), Suzgun and Irmen (14th–10th centuries BCE) cultures (Figure 2) were most likely proto-Ugric-speaking. This is again in line with our genetic model, in which the Mezhovskaya population was predicted to be the major constituent of the proto-Ugric gene pool.

The model of Grünthal et al., 2022 [4] implicitly upholds previous assumptions [32,33], that the Iron Age descendants of the aforementioned cultures: the Itkul (7th–3rd centuries BCE), Gorokhovo (6th–3rd centuries BCE), and Sargat (6th century BCE–4th century CE) (Figure 3), were most likely also Ugric-speaking. Despite the previous controversies and uncertainties surrounding the exact location of the Uralic homeland, there has been a consensus that the Proto-Ugric languages can be linked to the Bronze Age Mezhovskaya culture and the Iron Age Sargat-Gorokhovo cultures, primarily due to their geographical proximity to the present-day territories of the Mansi and Khanty languages, as well as their nomadic archaeological heritage. The presence of numerous equestrian terms in the Ugric languages indicated that Ugric speakers had a strong association with horsemanship in ancient times [33].

### 2.4. Consistency of Genetic and Archaeological Data

Archaeological data indicate that during the late Bronze Age, forest hunter Lozva-Atlym immigrants arrived in the Mezhovskaya territory from the northern Ob-Irtis region [34] (Figure 2). This finding closely parallels the genetically detected Mezhovskaya-proto-Nganasan admixture. Unfortunately, we currently lack DNA data from the Lozva-Atlym population to confirm this assumption. By the Iron Age the Lozva-Atlym descendants were already living among the members of the Mezhovskaya-descendant Itkul culture (Figure 3), and their archaeological remains are known as Gamayun [35], from which we also lack DNA data. Nonetheless, we have a single published Itkul genome (MJ-42) dated to 793–541 BCE [36] which indeed gave valid 3-source qpAdm models for Conq_Asia_Core, if supplemented with Sarmatian and East Asian sources (Maróti et al. unpublished), suggesting that the Itkul could be a potential Iron Age source population of Conq_Asia_Core.

Archaeological data indicate that in the 8th century BCE the Itkul bronze metalworkers suffered an even more determining impact from the south, as they were gradually integrated into the steppe societies of the Saka and Sarmatian tribes. This integration lead to the formation of the Gorokhovo culture (6th–3rd centuries BCE) (Figure 3), which was culturally and likely politically linked to the southern nomadic tribal confederations [35]. The early Sarmatians are associated with the Prokhorovka culture (500–300 BCE), which formed in the southern Urals, above the Aral and Caspian Seas, and was adjacent to the Tasmola culture (7th–3rd centuries BCE) of the Saka people (Figure 3). Surprisingly, these culturally related neighboring nomadic groups were genetically different, as the Sarmatian genomes harbored higher European components [37].

A similar integration happened east of the Gorokhovo, at the Tobol-Isim-Irtis region, where the Sargat culture (6th century BCE–4th century CE), the Iron Age successor of the Irmen, was fully integrated into the Scythian-Saka nomadic societies [35]. There are 18 published genomes from the Sargat material, which is genetically quite heterogeneous [37]. I would like to remind the reader once again that the Sargat people have been considered to be the ancestors of Hungarians [32,33,38] because of their nomadic grave goods and linguistically expected location. Indeed, the Sargat genome structure was the most similar to that of the Conq_Asia_Core in admixture analysis [5]. Nevertheless, each Sargat genome was rejected as a source in our qpAdm models, probably because they contained a small Ekven-Eskimo component [37], which was missing from Conq_Asia_Core. From these data we conclude that the Ugric-speaking Sargat population was closely related to the ancestors of Conq_Asia_Core, but it was not their direct ancestor. 

This conclusion also fits with the latest archaeological and anthropological indications. According to archaeological findings, anthropological types and geographical location, it is highly probable that the Gorokhovo culture is more closely related to the pre-Hungarian heritage than the Sargat culture [39]. Anthropological differences between the Sargat and Gorokhovo populations [40] align with genetic data. The Gorokhovo people migrated to the European side of the southern Urals during the 4th–3rd centuries BCE and merged into the early Sarmatian Prokhorovka culture [40,41], which is consistent with the roughly 40% Sarmatian admixture detected genetically in Conq_Asia_Core. While the genetic admixture was dated slightly earlier (643–431 BCE), the early Itkul–Sarmatian integration may explain this discrepancy. Overall, the archaeological, anthropological, and indirect genetic evidence supports the idea that the Gorokhovo culture is the most likely direct ancestor of Conq_Asia_Core. However, the lack of Gorokhovo DNA data currently prevents any conclusive genetic proof.

If the Gorokhovo culture can be traced back to the ancestors of the Conq_Asia_Core, it is likely that they experienced a significant cultural influence from the nomadic Sarmatians, who spoke an Iranian language. This influence is supported by the presence of Iranian loanwords in the Hungarian language [42]. Iranian cultural influences can also be identified in various cultural aspects, including jewelry, clothing adornments, burial customs, metalworking, and the distinct nomadic structure of the conquering Hungarian society. During the Sargat-Gorokhovo period, the archaeological record indicates the existence of a contiguous, massive Ugric-speaking block east of the Urals, living in complex hierarchical nomadic societies. The Gorokhovo and Sargat were interrelated, and the latter seem to have controlled the northern branch of the Silk Road [35], importing goods from Fergana, Bactria, Chorasmia, China and the Xiongnu world. At the height of their power in the 4th–2nd centuries BCE, the Sargat expanded westwards, incorporating the Gorokhovo territories, which could have triggered the Gorokhovo migration (Figure 3).

The archaeological record indicates that between the 2nd–4th centuries CE both the Sargat and Sarmatian cultures underwent a significant transformation due to the invading Xiongnus. Archaeologist Sergei Botalov proposed the existence of a Hunno-Sarmatian mixed culture near the Urals during this time [43,44], which is believed to be the forerunner of the European Huns. In this context it is relevant to note that we have detected genetic relations between the Xiongnus and European Huns, as well as Sarmatian genomes among the Huns [5]. Genetic studies have also demonstrated the presence of Xiongnu and European Hun populations in the Ural region. For example, the genome of the Kazakhstan_Nomad_Hun_Sarmatian individual, which was excavated in the southern Urals and dated to 330–530 CE, can be traced back to Mongolian Xiongnus [22]. Another sample, the Kurayly_Hun_380CE, was excavated southwest of the above, and belonged to a European Hun elite warrior [37]. His genome formed a genetic clade (was nearly identical) with that of our MSG-1 Hun elite warrior excavated in Transylvania, whose origin could also be traced back to the Xiongnus [5]. We dated the Conq_Asia_Core–Xiongnu/Hun admixture virtually to this period; furthermore, the Kazakhstan_Nomad_Hun_Sarmatian genome was a perfect source in the model [5], providing a sound fit between the genetic and archaeological data.

## 3. Implications

I have demonstrated that the interpretations of genetic, archaeological and linguistic data about the early Ugrians are reasonably consistent. Moreover, with this multidisciplinary approach I have been able to outline their prehistory roughly from the Bronze Age to late antiquity. However, the reconstruction of their genetic history from later periods is still hindered by the scarcity of comparative ancient DNA material. Medieval historical records depict dynamic changes on the Pontic-Caspian steppe, with the involvement of numerous peoples, mostly Turkic-speaking. In the following, I will investigate the possibility of Ugric speakers potentially appearing among them.

From the 4th–3rd centuries BCE, the Sarmatians spread from the southern Urals westward and southward [45] (Figure 3). By the beginning of the Common Era (CE), they held dominion over the Pontic-Caspian Steppe, and starting from 50 CE, they exerted control over the Great Hungarian Plain until the arrival of the Hun invasions. The genetically detected early-Sarmatian admixture in Conq_Asia_Core raises the possibility that some groups of proto-Hungarians might have moved together with their Iranian-speaking associates. In light of the Hun genetic admixture detected in Conq_Asia_Core, proto-Hungarian groups could similarly have been swept away by the westward-migrating Huns. However, the westward migration of the Xiongnus first impacted the Sargat population from the 1st–2nd centuries CE, whereas they reached the proto-Hungarians somewhat later. This is accurately reflected in the archaeological evidence and is well supported by our genetic data as well. Archaeologically, the initial influences of the Xiongnu within the Sargat culture can be dated back to the 2nd century on the Siberian side of the Urals. However, most of the late Sargat discoveries originate from the western side of the Urals, indicating evidence of depopulation and political disintegration [35]. This 3rd–4th century CE westward movement and decline was most likely the direct consequence of the Xiongnu migrations. We have similarly dated the appearance of the Hunnic admixture within the Conq_Asia_Core to the 3rd century CE. 

A significant portion of the Hunnic migrations, moving westward from Mongolia, most likely followed the northernmost branch of the Silk Road. This route largely coincides with the expansion path of the Bronze Age Seima-Turbino culture; furthermore, this route served as a major migratory pathway in later periods as well [35]. It seems feasible that on their way, the Xiongnus integrated a significant part of the Ugric-speaking communities. As a result, these communities left their homes and moved together with their new masters. Therefore, Ugrian populations were very likely part of the medieval course of events on the Pontic-Caspian steppes, listed in historical sources, right from the Hun period. There is an approximately 200-year-long gap between the disappearance of the Xiongnus (155 CE) and the appearance of European Huns (370 CE) in historical sources. In the absence of conclusive evidence, the Xiongnu-Hun relations have been debated [46]. However, both the genetic data and the archaeological record provide growing evidence that during this period the Xiongnus resided between the Altai and the Urals, mostly in Ugric territories. By the time they crossed the Volga in 370 CE, their power was significantly augmented by the recruited Ugric and Sarmatian peoples. 

Genomic analysis of the populations mentioned in the article, which have not yet been genetically examined, along with additional genomic studies focusing on relevant archaeological cultures from the Ural-Altaic region, will significantly contribute to a more precise understanding of the prehistory of Ugric-speaking peoples.

## Figures and Tables

**Figure 1 genes-14-01345-f001:**
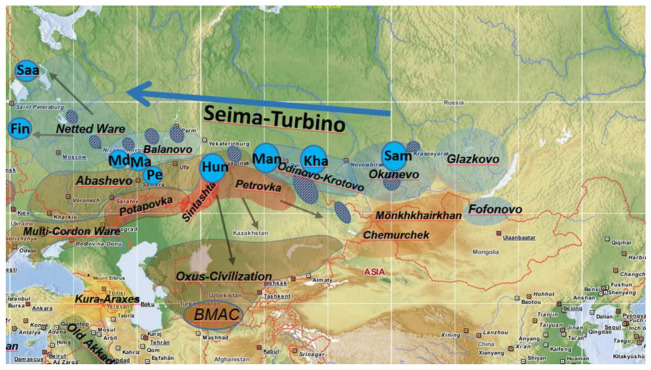
The relevant Eurasian archaeological cultures in the early Bronze Age (ca. 2300–1700 BCE) with their approximate ranges. Chequered ovals: Seima-Turbino major sites. Labeled blue ovals: core locations of Uralic branch ancestors: Saa(mi), Fin(nic), M(or)d(vin), Ma(ri), Pe(rmic), Hun(garian), Man(si), Kha(nty), and Sam(oyedic). Redrawn from Grünthal et al., 2022 [4].

**Figure 2 genes-14-01345-f002:**
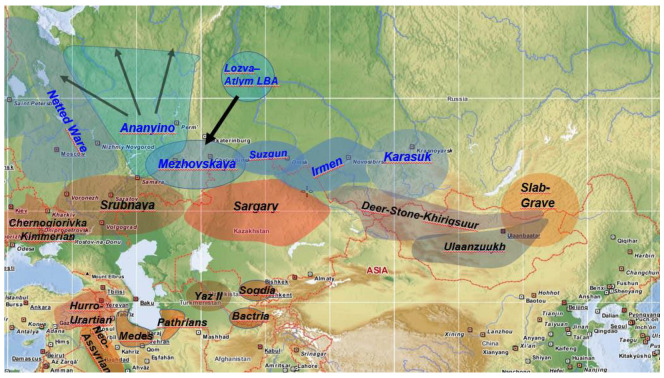
The relevant Eurasian archaeological cultures in the late Bronze Age (ca. 1500–750 BCE) with their approximate ranges. Cultures in the Seima-Turbino zone are indicated with blue letters.

**Figure 3 genes-14-01345-f003:**
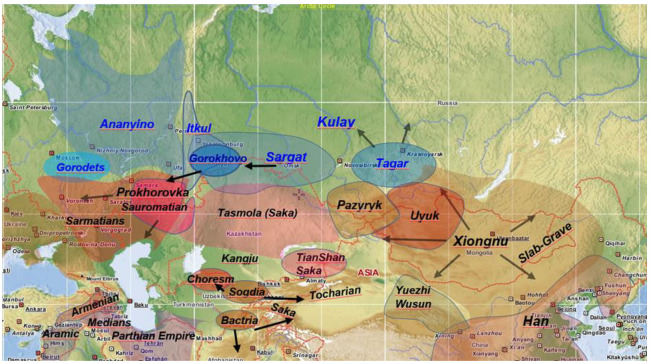
The relevant Eurasian archaeological cultures in the Iron Age (ca. 800–100 BCE) with their approximate ranges. Cultures in the Seima-Turbino zone are indicated with blue letters.

## Data Availability

No new data were created.

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
