# Peer review of "Integrating Linguistic, Archaeological and Genetic Perspectives Unfold the Origin of Ugrians"

_genes, 2023, doi:10.3390/genes14071345_

Round 1
Reviewer 1 Report
Interestingly, the Hungarians are linguistically isolated in Europe, even though they are genetically close to the surrounding populations. Although there are/were not genetic differences between human populations (in present or in the past) that could be explained culturally or linguistically, the manuscript can be considered as an attempt to describe a gradual development of Ugric-speaking ethnicities since the Bronze Age to Antiquity and to provide some insight to latest evolutionary aspects of Central Europe. However, it does not provide to the reader an intelligible text. I read it several times and it did not bring much to me. I think the manuscript needs substantial revision.
I have some comments that might help.
The manuscript uses too many terms of archaeological cultures, which would be rather confusing for readership not versed in Bronze Age of Central and Eastern archaeology. Try to limit these local terms and be more general in your description. Use rather some qpGraphs to describe the genetic relationships of the aDNA samples.
Description of the previous (before Bronze Age) population history of the Carpathian Basin is missing. I think just 2-3 paragraphs describing what happened before Bronze Age especially in the Neolithic might be mentioned too. Contemporary population structure of Europe was completed at this time. For example, it seems that steppe element (Yamnaya) penetrated to eastern Europe at Late Neolithic and it is present in Hungary today in a higher proportion than in Ukraine or Belorussia (see for example Haak et al. 2015. Nature 522:207-11). What happened with this component during the time of Bronze Age?
I think that correlation between genetic and archaeological data is just a speculation. Why the Seima-Turbino complex should be considered as Uralic-speaking? How such connection could be scientifically proven? Please consider this issue in the chapter 2.2.
The author write that Siberian component „was diluted to a nearly undetectable level“ - in fact, what is nearly undetectable level? Can it be specified better? Is that component detected in a current Hungarians or is not?
Reviewer 2 Report
The genesis of the Uralic-speaking populations, as well as the origin of these groups, is a scientific problem addressed by specialists from various scientific fields. The manuscript shows the impressive results of the connection of several sources obtained by the methods of paleogenetics and linguistics in the context of archaeology, and partly biological anthropology. The author of the manuscript carefully selected a model that is in good agreement with the data of linguistics and genetics, and at the same time does not contradict the archaeological source. Note that the attached graphic schemes make it possible to understand the complex names of archaeological cultures, and make the perception of the author's ideas optimally accessible. I was pleased to see in this model the confirmation of the ideas of anthropologists, in particular, of Alexander Kozintsev, who convincingly showed the protomorphic nature of the Ural anthropological complex 20 years ago based on the results of a craniological study. Uralic-speaking populations have specific morphological features that cannot be explained by anything other than a common descent from a protomorphic group. The origins and range of this ancient group caused great discussions among anthropologists and archaeologists, and the appeal to the results of the linguistic research did not give any serious arguments for stopping the discussion. Today, paleogenetic and linguistic data in the context of archaeology can help to find objective reconstructions of bio-social processes in the histories of the Ural-speaking peoples. The presented manuscript is an example of such a successful interdisciplinary reconstruction.
Round 2
Reviewer 1 Report
The Author addressed the issues and the manuscript is ready to be published.